# Post-Polio Syndrome Revisited

Michael Punsoni, Nelli S. Lakis, Michelle Mellion and Suzanne M. de la Monte *

Division of Neuropathology, Departments of Pathology and Laboratory Medicine, Neurology, and Neurosurgery, Rhode Island Hospital, Alpert Medical School of Brown University, Providence, RI 02903, USA; mpunsoni@gmail.com (M.P.); nlakis@kumc.edu (N.S.L.); mmellionmd@gmail.com (M.M.)

* Correspondence: suzanne_delamonte_md@brown.edu; Tel.: +1-401-444-7364; Fax: 401-444-2939

**Abstract:** Post-polio syndrome (PPS) is characterized by recrudescence or worsening of motor neuron disease symptoms decades after recovery from acute paralytic poliovirus infection, i.e., poliomyelitis. PPS afflicts between 25% and 40% of poliomyelitis survivors and mimics motor neuron diseases (MNDs), such as amyotrophic lateral sclerosis (ALS), due to its selective impairment, degeneration, or death of motor neurons in the brainstem and spinal cord. Herein, we report a case of PPS in a 68-year-old man with a remote history of bulbar and cervical cord involvement by poliomyelitis, review the relevant literature, and contrast the salient histopathologic features that distinguish our case of PPS from ALS.

**Keywords:** post-polio syndrome; motor neuron disease; amyotrophic lateral sclerosis





## 1. Introduction

Poliomyelitis is a viral infectious disease caused by any one of the three poliovirus strains, although Type 1 is the most frequently identified pathogen [1]. In the United States, the use of poliovirus vaccination drastically reduced the annual incidence rate from over 10,000 cases to about 10, and virtually eradicated wild poliovirus [1]. On a global scale, polio immunizations reduced acute paralytic disease from over 600,000 cases per year prior to the campaign to fewer than 1000 by the year 2000 [2]. However, low endemic rates persist in global pockets due to poor access or compliance with vaccination [2–4].

Following acute infection and viremia, in about 1 percent of cases, poliovirus spreads to the central nervous system (CNS). Poliovirus preferentially replicates in and destroys motor neurons of the brainstem, spinal cord, and motor cortex, causing paralytic poliomyelitis [5–7]. Accompanying neurological signs and symptoms include asymmetric weakness in various muscles, difficulty swallowing, myalgias, loss of superficial and deep tendon reflexes, and problems with bowel and bladder function [7]. The involvement of brainstem structures is most critical due to inflammation and destruction of neurons that regulate respiration. Supportive care to cover the life-threatening phases of acute primary CNS infection enables recovery, which can be partial or complete. Degrees of recovery vary with age at onset, the extent of CNS involvement, availability of supportive care, and proneness to complications. Younger individuals are more capable of recovery than older individuals due to their inherently greater potential for plasticity and repair [1,2].

Decades after the polio epidemic, it was discovered that between 25% and 40% of people who had recovered either fully or partially from poliomyelitis developed motor neuron disease with features that were indistinguishable from sporadic motor neuron disease (MND) [8,9]. Drawing connections with past histories of poliomyelitis was challenging due to the perception that the later life-presenting symptoms corresponded to a new disease entity if recovery from the childhood illness had been complete, whereas if recovery had been incomplete, the motor dysfunction likely represented an exacerbation of pre-existing poliomyelitis-related pathology. Eventually, the two concepts were fused by consensus and are now termed 'post-polio syndrome' (PPS) [10–14]. PPS uniquely differs from sporadic

or primary MND due to its natural history corresponding to what is regarded as a clinical recrudescence of a childhood poliovirus CNS infection.

The three main clusters of MND, amyotrophic lateral sclerosis (ALS), spinal muscular atrophy, and primary lateral sclerosis, have sporadic and genetic occurrences. Survival durations are substantially abbreviated compared with other forms of neurodegeneration and range from 1 to 5 years [15–18] compared with 7 to 20 years for Parkinson's, Alzheimer's disease (AD), dementia with Lewy bodies, and frontal-temporal lobar degeneration [19–23]. Advances in human postmortem neuropathologic research have demonstrated significant overlap between MND and dementia-associated neurodegenerative diseases [24], which could be impactful in shortening survival [21].

PPS mimics MND due to its relatively selective impairment, degeneration, or death of motor neurons in the spinal cord and brainstem. The mechanism of PPS is not well understood. Potential etiologies include: (1) re-awakening of underlying pathologies that had been masked by repair and recovery but rendered weakened due to aging-associated neuronal and fiber attrition; and (2) prior poliomyelitis causes epigenetic changes that increase vulnerability to sporadic MND. These concepts could potentially be addressed by comparing the molecular pathologies in PPS and MND, particularly spinal muscular atrophy.

## 2. Report of a Case

A 68-year-old White man was evaluated at the Rhode Island Hospital for weakness in both upper extremities that had progressed slowly over the previous several years. In addition, the patient suffered from chronic respiratory insufficiency, anxiety, asthma, and hypertension. The patient had a remote past medical history of bulbar and cervical cord poliomyelitis at 2 years of age that mostly resolved except for non-progressive difficulty lifting his arms. Worsening motor weakness began over 30 years after stable recovery from poliomyelitis. A neurological examination late in the clinical course documenting muscle atrophy, and electromyography showing denervation, led to the diagnosis of PPS. His two female siblings reported no symptoms corresponding to MND. Molecular studies to exclude specific gene abnormalities linked to ALS, SMA or primary lateral sclerosis were not performed. The case was independently reviewed by a second neurologist who confirmed the likely clinical diagnosis of PPS. Throughout his course, the patient was non-compliant with treatment recommendations. He died from an acute myocardial infarct in the setting of severe coronary atherosclerosis affecting all major vessels, mitral, aortic, and tricuspid valve calcification and stenosis, hypertensive cardiomyopathy, pulmonary hypertension, and congestive heart failure. Consent was granted for a full postmortem examination.

Macroscopic examination of the brain fixed in 10% neutral buffered formalin revealed diffuse, symmetrical cerebral edema with impending bi-uncal herniation (brain weight 1465 g), severe calcific, non-occlusive atherosclerosis of the internal carotid artery and all major posterior circulation vessels linked to the circle of Willis, and atrophy of the frontal and temporal lobes. Paraffin-embedded histological sections prepared from 20 standardized regions of brain and spinal cord, were stained with Luxol fast blue-hematoxylin and eosin (LHE). Additional sections from selected blocks were stained by Bielschowsky silver impregnation, or immunostained with antibodies to phospho-tau, neurofilament, ubiquitin, glial fibrillary acidic protein, alpha-synuclein, and CD68 to detect neurodegeneration and tract degeneration. Sections of quadriceps and gastrocnemius muscles were stained with H&E and immunostained with antibodies to slow and fast myosin. For immunohistochemistry, 4-microns-thick paraffin sections were mounted onto Leica Surgipath + coated slides. The entire immunostaining procedure was performed using a DAKO Autostainer Plus automated staining system.

The findings related to PPS consisted of striking asymmetric atrophy, neuronal loss, and gliosis in the ventral horns at all levels of the spinal cord, but most prominently in the cervical and thoracic cord (Figure 1A–C). Neuronal inclusions corresponding to Bunina

bodies were not detected, including by immunohistochemical staining with antibodies to ubiquitin (Figure 1D) and TAR DNA-binding protein 43 (TDP-43) (not shown). Immunostaining for neurofilament confirmed the loss of spinal anterior horn cells. Correspondingly, ventral and cauda equina nerve roots exhibited up to 50% losses of myelin (Figure 2A) and axons (Figure 2B) by LHE and neurofilament immunohistochemistry (or Bielschowsky staining), respectively. Regions with extensive fiber loss exhibited replacement fibrosis. In contrast, there was no evidence of motor neuron loss in the precentral gyrus or brainstem motor nuclei, and loss of myelinated fibers in corticospinal tracts was modest to nil. The primary motor cortex and pyramidal tracts were intact bilaterally. Sections of quadriceps and gastrocnemius muscles showed modest atrophy but conspicuous myofiber type grouping by immunohistochemical staining of slow and fast myosin (Figure 2C,D).

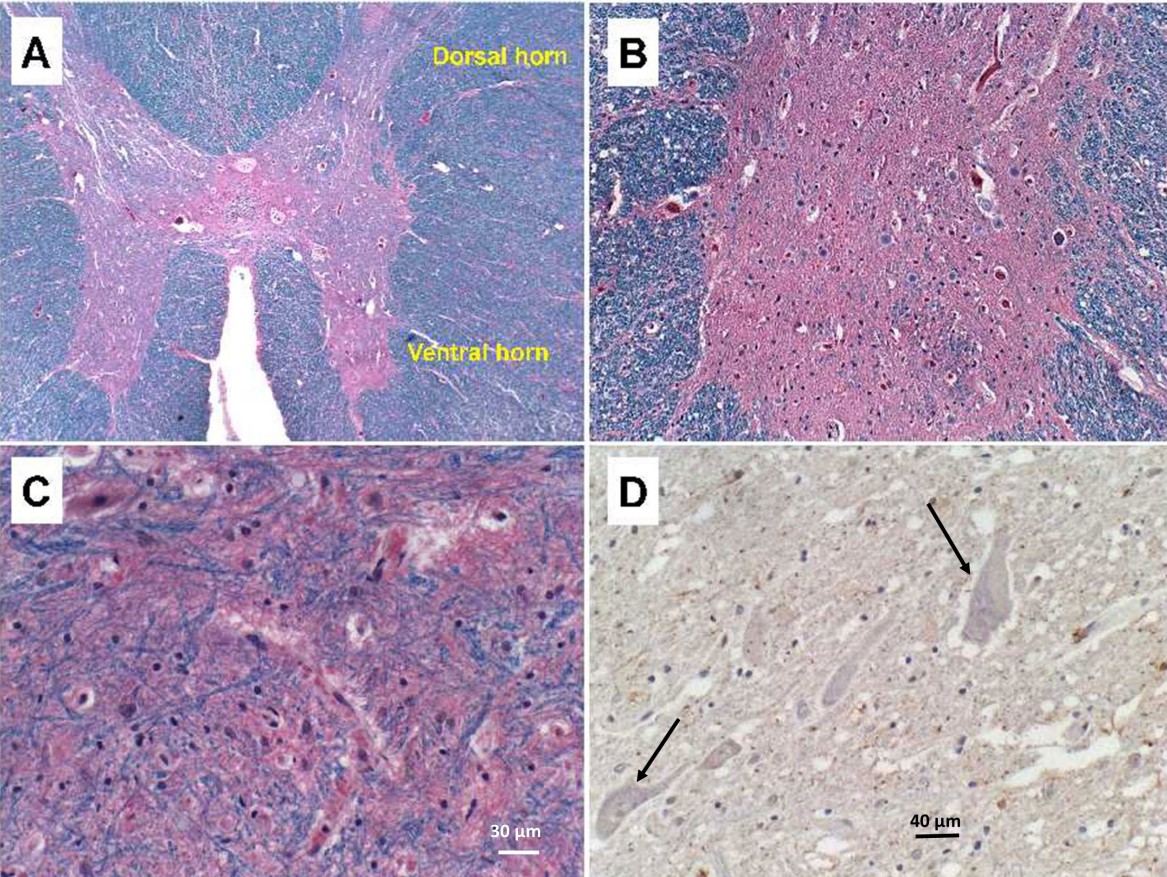

**Figure 1.** Spinal ventral horn atrophy with neurodegeneration in PPS. Formalin-fixed, paraffin-embedded histological sections of thoracic spinal cord were (**A**–**C**) stained with Luxol fast blue, hematoxylin and eosin (LHE; 8 μm thick), or (**D**) immunostained for ubiquitin (4 μm thick). (**A**) Thoracic cord level showing absence of lateral and anterior corticospinal tract degeneration (note homogeneous Luxol fast blue staining of myelin). (**B**) Extensive neuronal loss in the ventral horn shown in Panel A. (**C**) Ventral horn cells are virtually replaced by glia. (**D**) Note the absence of ubiquitin-positive inclusions in residual anterior horn cells (examples of negative staining in neurons shown at arrow tips). (**C**) Scale bar = 30 μm; (**D**) Scale bar = 40 μm.

Due to the presence of frontal and temporal lobe atrophy on macroscopic examination together with the age of the patient, sections of anterior frontal, posterior frontal, parietal and occipital lobes including cortex and underlying white matter, and the amygdala were immunostained for phospho-tau, ubiquitin, and amyloid-beta. In addition, sections of anterior frontal, hippocampus, and amygdala were stained by Bielschowsky silver impregnation. Bielschowsky staining demonstrated scattered neurofibrillary tangles

in the anterior frontal cortex, and moderate densities of neurofibrillary tangles, neuritic and diffuse plaques, and dystrophic neurites in the entorhinal cortex, CA1–CA2 of the hippocampal formation, and ventromedial amygdala. Phospho-tau and ubiquitin immuno-histochemical staining detected those same abnormalities, together with abundant labeling of white matter glial cells. Amyloid-beta immunostaining demonstrated moderate densities of senile plaques and amyloid angiopathy, which were in low abundance in the frontal lobe, but moderate in the parietal, occipital and temporal cortex and in the amygdala. These findings correspond with early-stage AD, Braak Stage III/VI.

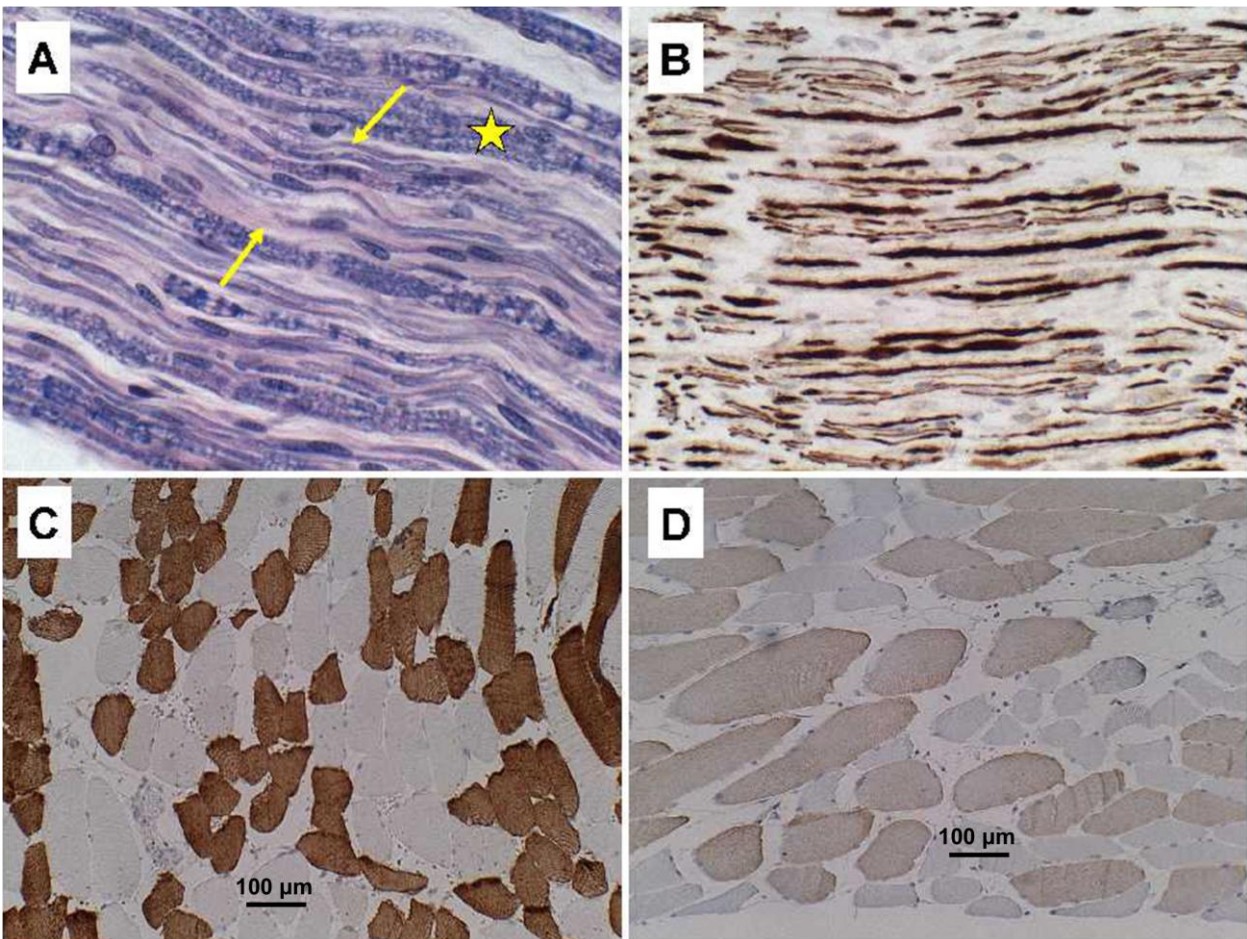

**Figure 2.** Ventral nerve root degeneration and skeletal muscle myofiber type grouping in PPS. Spinal ventral nerve roots (**A**) stained with LHE or (**B**) immunostained with antibodies to neurofilament show loss of myelinated fibers in A. For example, between arrows, asterisk marks normally myelinated fibers, and in (**B**), loss of axon is represented by the clear gaps among fibers. (**C,D**) Histological sections of quadriceps muscle immunostained to detect (**C**) fast or (**D**) slow myosin. Note clear grouping according to myofiber type, i.e., fast versus slow. Brown staining precipitates in (**B**–**D**) correspond to positive immunostaining results. (**C,D**) Scale bars = 100 µm.

Additional histopathological findings included, multiple micro-infarcts in the occipital lobes and thalamus, hypoxic-ischemic leukoencephalopathy in the occipital lobes, and cerebral edema.

## 3. Review of Cases

Due to the presence of overlapping early AD-type neurodegeneration, we considered ALS in the differential diagnosis. To help solidify the diagnosis of PPS, we compared the findings in our index case to six ALS cases that were autopsied within the same time period.

The ALS cases ranged from 60 to 80 years of age. All were male with White race, and none had a history of poliomyelitis. The cases were donated to the Brown University Brain Bank (re-named Brain Tissue Resource Center at Brown University) for research. Oversight for the human postmortem studies was provided by the Lifespan IRB committee #008303. The approval code was Human Studies Exemption #211037, initially issued on 30 March 2003. The postmortem brains, spinal cords, and skeletal muscle specimens were processed and evaluated using the same standard protocol. In all six cases, the populations of large motor neurons in Brodmann Area 4 (precentral gyrus) were markedly diminished. Loss of neurons was associated with gliosis of deep cortical layers and subcortical white matter, and Wallerian degeneration accompanied by axonal spheroids in the lateral and anterior corticospinal tracts at all levels (Supplementary Figure S1). Atrophy of the spinal anterior horns was associated with neuronal degeneration, neuronal loss, replacement gliosis (Figure 3A–C) and neuronal cytoplasmic inclusion bodies that were readily detected in LHE-stained sections (Figure 3D). Immunohistochemical staining confirmed the inclusion bodies corresponded to TDP-43-positive (Figure 3E) and ubiquitin-positive (Figure 3F) Bunina bodies. All six cases of ALS had variable degrees of overlapping neurodegeneration, e.g., FTLD, PD, or AD.

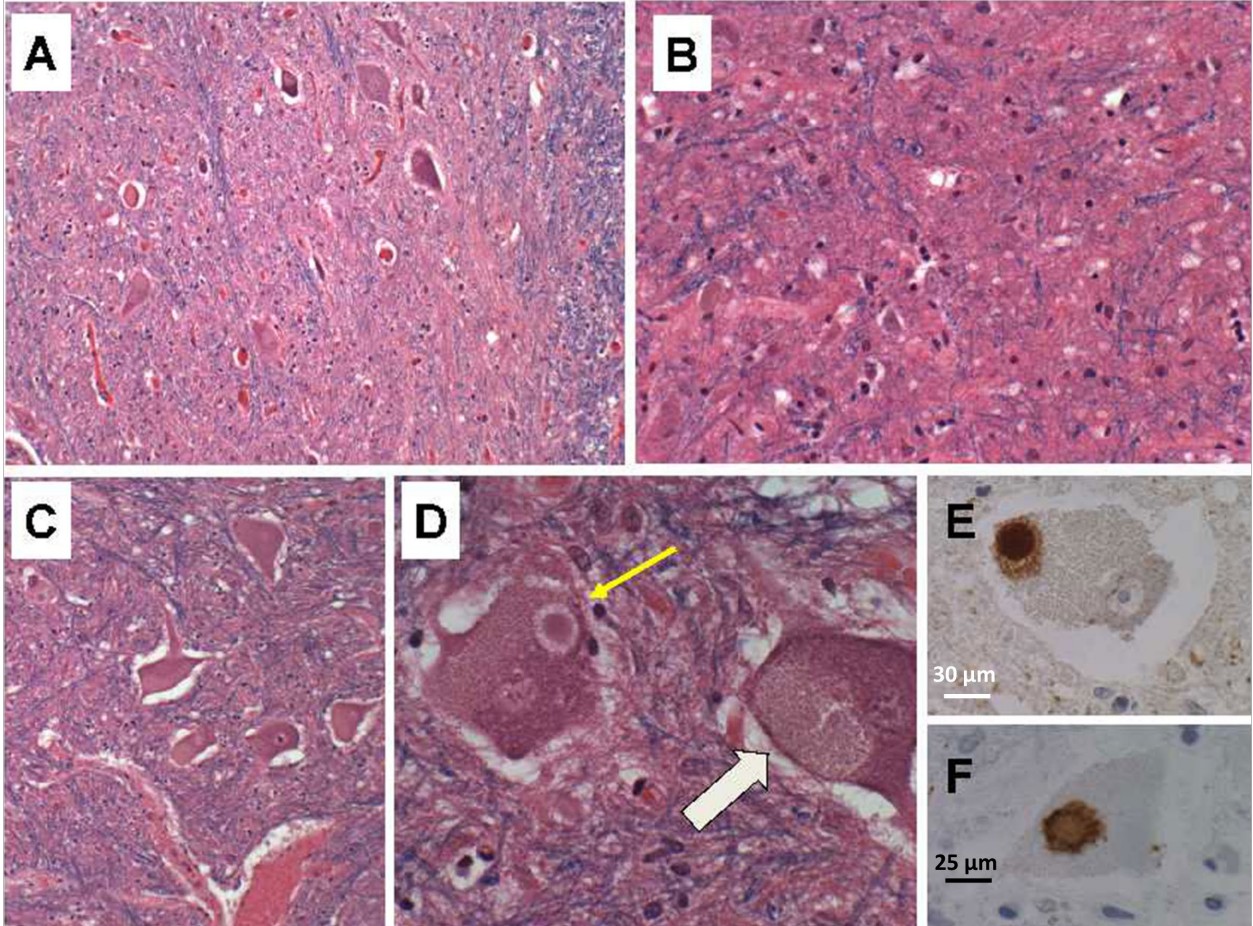

**Figure 3.** Ventral horn neurodegeneration in ALS. Spinal thoracic ventral horn showing (**A**) subtotal loss of neurons and (**B**) extensive gliosis. (**C**) Many surviving anterior horn cells exhibit swelling and chromatolysis. (**D**) Bunina body cytoplasmic inclusions. The narrow arrow shows a classical Bunina body. The broad arrow shows a larger, diffusely pale cytoplasmic inclusion. Additional sections from the same block were immunostained with antibodies to (**E**) ubiquitin and (**F**) TDP-43, both of which detected discrete cytoplasmic Bunina bodies (dense brown staining). (**E**) Scale bar = 30 μm; (**F**) Scale bar = 25 μm.

Ventral nerve root degeneration in the ALS cases was associated with loss of myelin and axons with notably greater severities in lumbar compared with thoracic levels (Figure 4). LHE stains highlighted myelin loss, vacuolation, thinning fiber (Figure 4A,B). Neurofilament immunohistochemical staining revealed conspicuously greater degrees of axonal loss, axonal swelling, and fiber variability in lumbar (Figure 4C) compared with thoracic (Figure 4D) nerve roots.

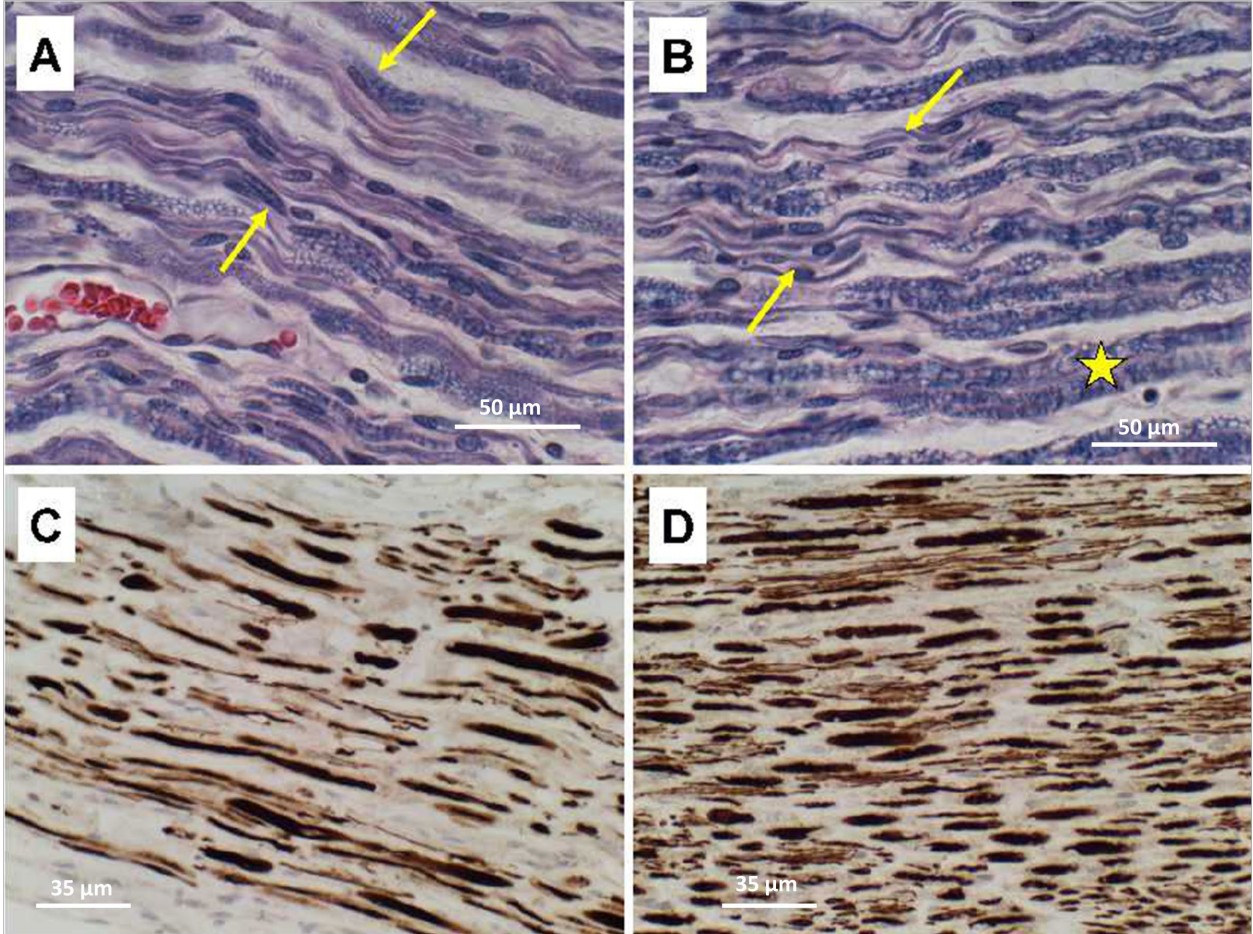

**Figure 4.** Spinal cord (**A,C**) Lumbar and (**B,D**) thoracic ventral nerve root degeneration in ALS. Formalin-fixed paraffin-embedded LHE-stained sections in (**A,B**) show loss of myelinated fibers, e.g., between arrows (asterisk marks normally myelinated fibers). (**C,D**) Additional sections from the same blocks immunostained with antibodies to neurofilament to show loss of axon (gaps among fibers). Fiber degeneration and loss are greater in lumbar (**A,C**) compared with thoracic (**B,D**) cord levels. (**A,B**) Scale bars = 50 µm; (**C,D**) Scale bars = 35 µm.

In contrast to the PPS case, quadriceps skeletal muscle samples from the ALS cases showed regions of denervation atrophy characterized by the presence of both small (Figure 5A) and large (Figure 5B) grouped atrophy along with myofiber hypertrophy. Immunohistochemical staining for fast (Type 2) (Figure 5C) and slow (Type 1) (Figure 5D) myosin revealed clear myofiber type grouping in less atrophic regions, similar to the findings in PPS (see Figure 2).

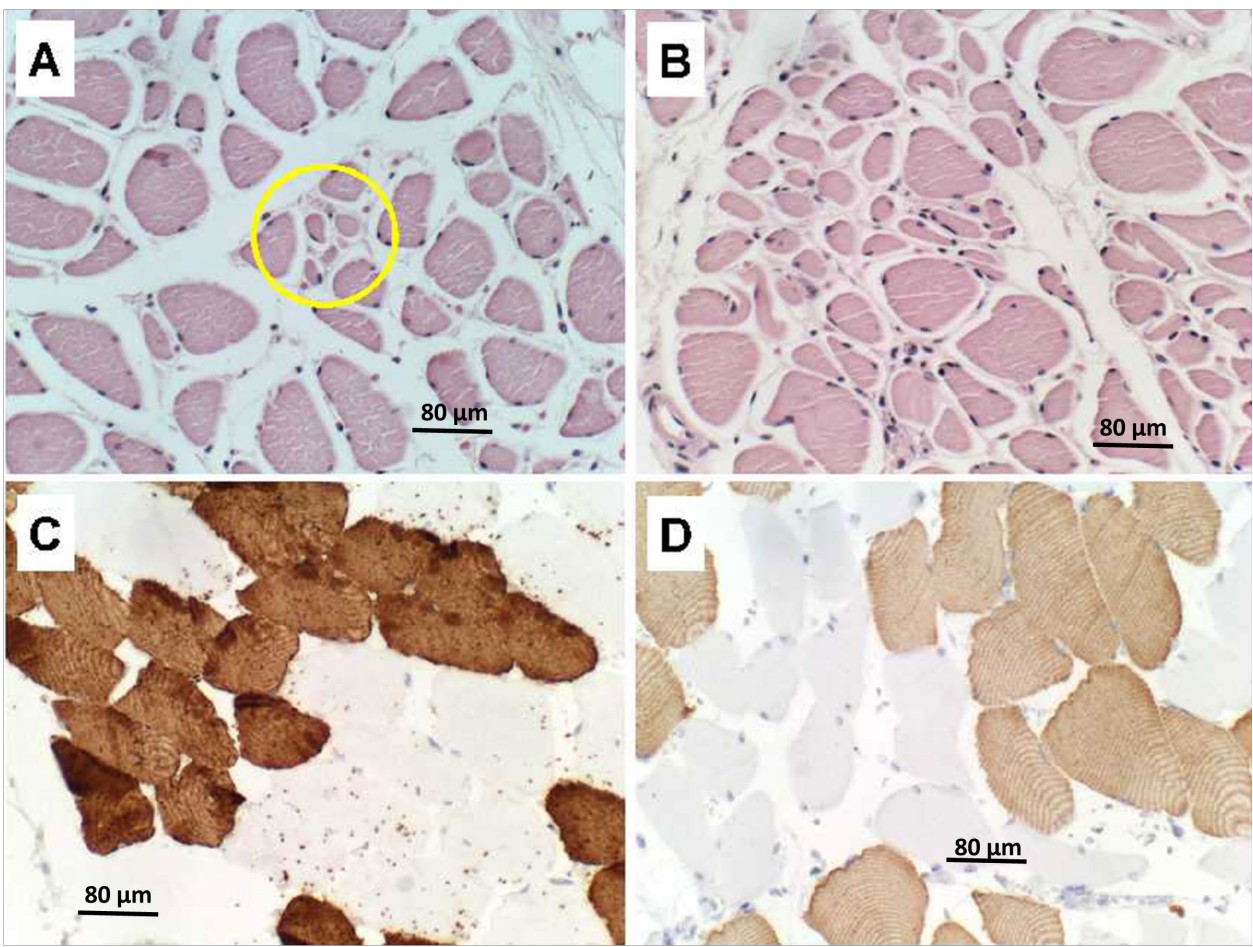

**Figure 5.** Skeletal muscle (**A**,**B**) denervation atrophy with myofiber hypertrophy and myofiber type grouping in ALS. Histological sections stained with H&E show (**A**) small (circled) and (**B**) large group myofiber atrophy with no endomesial fibrosis in different areas of quadriceps muscle. Additional sections from the same blocks immunostained to detect (**C**) fast (Type 2) or (**D**) slow (Type 1) myosin show clear myofiber type grouping away from the regions with active denervation as shown in Panels (**A**,**B**). (**A**–**D**) Scale bars = 80 μm.

### 4. Discussion

The term post-polio syndrome (PPS) was initially coined as an attempt to describe the multitude of neurological symptoms and functional impairments encountered by long-term survivors of poliomyelitis [25–29]. PPS, defined as "the development of new muscle weakness and fatigue in skeletal or bulbar muscles, beginning 15–30 years after an acute attack of paralytic poliomyelitis", afflicts 25% to 40% of the cases [30]. Although it is widely accepted that PPS is causally linked to an earlier bout of poliomyelitis, the mechanism driving the recrudescence of motor neuron disease symptoms and neurodegeneration remains unknown. One potential consideration is that compensatory responses leading to recovery from poliomyelitis or stabilization of symptoms decompensate over time, particularly with aging.

Clinically, the pre-morbid states that characterize patients who develop PPS include: (1) those who experience prolonged periods (up to several decades) of stable normal motor function; and (2) individuals with chronic fatigue and muscle weakness. The former is mainly associated with recovery from childhood poliomyelitis. The latter most likely reflects incomplete recovery from poliomyelitis, corresponding with our index case. In those individuals, PPS is associated with both worsening of baseline symptoms and the emergence of new signs and symptoms. Limbs that were originally affected tend to be

involved, although there are exceptions. A diagnosis of PPS in people who experience residual chronic fatigue and muscle weakness can be challenging because the signs and symptoms overlap with chronic fatigue syndrome [31].

The slowness of muscle weakness progression in PPS is partly due to the relatively focal involvement of motor units within multiple muscles. Declines in motor function caused by worsening of weakness or loss of strength in previously uninvolved muscles compromise the ability to carry out normal daily activities. Normal functions can be further impaired by accompanying muscle pain, difficulties with breathing and swallowing, and reduced tolerance to cold [32,33]. The perception that increased physical activity can enhance recovery and strength following poliomyelitis is probably incorrect because instead it can worsen PPS symptoms and the severity of disability. Progressive weakness arising in muscles that were not previously affected by poliomyelitis or associated with persistent chronic fatigue and weakness could reflect a different pathophysiological process causing de novo degeneration of lower motor neurons. To distinguish it from the classical PPS, this entity has been termed "progressive post-poliomyelitis muscular atrophy (PMA)" [26].

The histopathological features of PPS are not well established compared with ALS and other MNDs due to the relatively uncommon nature of disease and overall decline in autopsy rates [34]. From the relatively few examples in the published literature, together with our own index postmortem case, the common targets of idiopathic/genetic MND and PPS include, most spinal anterior horn cells (most common) and brainstem cranial nerve motor nuclei, with attendant denervation myopathy involving affected motor units. In contrast to ALS, PPS seldom affects neurons in the brain. However, there are unusual reports of PPS associated with neurodegeneration in the precentral gyrus, corpus striatum, cerebellar cortex, and basal pontine nuclei [35]. Correspondingly, beyond the classical presentation of limb weakness, involvement of extra-spinal cord structures in the central nervous system impairs motor function in the face, pharynx, and larynx, contributing to difficulties with swallowing and breathing. Motor cortex neurodegeneration can lead to spasticity, and extrapyramidal motor system involvement causes problems with coordination and balance, further compromising independence, mobility, and ability to carry out activities of daily living.

The extraordinarily high rate of PPS among survivors of acute poliomyelitis speaks volumes for factors related to the initial infection or mechanisms of recovery as key pathogenic mediators of PPS-associated neurodegeneration. One popular theory known as "neural fatigue" infers that acute poliovirus infections target and kill specific spinal anterior horn cells and brainstem motor nuclei. Functional recovery is mediated by the neuritic sprouting of surviving motor neurons with attendant re-innervation of denervated skeletal muscle. The degrees to which re-innervation is effective may dictate resultant levels of strength. However, re-innervation sprouting leads to the enlargement of motor units, adding metabolic stress to remaining motor neurons. Chronically increased metabolic stress could eventually trigger a second wave of neurodegeneration and disability, corresponding with the emergence of PPS. Furthermore, it is important to bear in mind that within the 15- to 30-year interval between acute poliomyelitis and PPS-associated loss of motor neurons, aging occurs in the patients. Aging itself leads to oxidative and metabolic stress via impairments in insulin signaling and mitochondrial dysfunction [36–38]. Therefore, intrinsic neuronal stress related to recovery-induced expansion of the motor unit plus aging, could account for the development of PPS in long-term survivors of poliomyelitis. Similarly, perhaps the increased oxidative stress induced by exercise accounts for the worsening of PPS symptoms with physical activity. At the same time, these concepts suggest that agents to enhance efficiency of metabolic function and reduce oxidative and mitochondrial stress would provide therapeutic remediation for PPS.

A final consideration is whether PPS is actually a form of MND, including ALS or progressive muscular atrophy (PMA). This notion is supported by the overlapping symptoms in PPS and ALS or PMA. In particular, PMA, the rapidly progressive form of PPS, mimics the tempo of ALS, and a few cases of PPS have been shown to progress

to ALS [39,40]. Furthermore, the neuropathology of spinal and brainstem motor neuron degeneration in PPS is quite like PMA and ALS, except for the absence of Bunina bodies in PPS and their presence in ALS. On the other hand, with extensive neuronal loss, Bunina bodies would not be detected, rendering the definitive diagnosis difficult in the absence of an ample clinical history. The differential diagnosis is further clouded by the finding that like ALS, PPS can also overlap with other neurodegenerative diseases, as demonstrated by the early AD pathology detected in our index case. Therefore, consideration should be given to the potential etiopathogenic relatedness of PPS and sporadic forms of ALS or PMA, perhaps from the perspective of how other agents, exposures, toxins, etc., might mimic poliovirus-mediated damage to CNS motor neurons.

**Supplementary Materials:** The following supporting information can be downloaded at: https://www.mdpi.com/article/10.3390/neurolint15020035/s1, Supplementary Figure S1: Corticospinal tract degeneration in ALS. Formalin fixed, paraffin embedded histological sections of (A,B) cervical or (C,D) thoracic spinal cord stained with Luxol fast blue, hematoxylin and eosin (LHE) revealed various degrees of tract degeneration in the lateral columns (LC) ranging from (A) long established with dense gliosis and myelin pallor, to (B,C) moderate but extensive with clear but less well-delineated regions of myelin pallor and vacuolation, to (D) mild with subtle myelin pallor but prominent vacuolation. Compare regions of myelin pallor (pink) with the denser luxol fast blue staining in the posterior columns. Ventral horns (VH) are atrophic, but in Panel B, the degeneration is quite severe, resulting in a blunted appearance.

**Author Contributions:** M.P., data curation, formal analysis of the case, initial writing of the manuscript, case investigations, and methodology; N.S.L., data curation, formal analysis, investigation; M.M., formal analysis, investigation, resources, validation; S.M.d.l.M., data curation, formal analysis, funding, investigation, methodology, supervision, writing, writing and editing, visualization, project administration. All authors have read and agreed to the published version of the manuscript.

**Funding:** This research was funded by grants from the National Institutes of Health-National Institute of Alcohol Abuse and Alcoholism: AA-011431 and AA-024018.

**Institutional Review Board Statement:** Oversight for the human postmortem studies was provided by the Lifespan IRB committee #008303. The approval code was Human Studies Exemption #211037, initially issued on 30 March 2003.

**Informed Consent Statement:** Patient consent was waived due to the patients were all deceased. And we obtained a autopsy authorization which is our standard form that grants consent for the use of decedents' samples and case information for education, training, research, and publication.

**Data Availability Statement:** The case report information and images are available upon request from the corresponding author: Suzanne_DeLaMonte_MD@Brown.edu.

**Conflicts of Interest:** All authors declare to have no conflict of interest related to this manuscript.

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
