# Peer review of "Post-Polio Syndrome Revisited"

_2035-8377, doi:10.3390/neurolint15020035_

Round 1

Reviewer 1 Report

In this manuscript, the authors presented a case report that a 68-year-old man got post-polio syndrome (PPS) diagnosis. Through autopsy, they found that the neuropathology of spinal cord and brainstem motor neuron degeneration in PPS is very similar to that of PMA and ALS, and overlaps with the pathology of other neurodegenerative diseases such as early AD, so PPS may be considered in the pathogenesis of sporadic ALS or PMA on the correlation. The conclusions can support the authors' observations and inferences in this regard, and have a novel explanation for the pathological causes of PPS. Although the authors' descriptions on this aspect are relatively complete, I still have some minor comments for their reference.

1. Because many MNDs have a genetic background, the patient's race must be mentioned. In addition, the age of the patient's previous poliomyelitis should also be provided to help clarify the possible relationship between PPS and MND.

2. If possible, since the authors declared that the patient has been diagnosed with PPS, the relevant specific symptoms, medical history and symptoms, and even the treatment method should be clearly described.

3. Has the patient ever received relevant genetic background checks of some specific genes (e.g. SOD1, C9ORF72, TARDBP, FUS, SMN1, ALS2 etc.) to clearly rule out the possibility of suffering from ALS, SMA, and PLS? This is especially important since the authors mention that PPS may actually be a form of MND.

4. All IHC images are suggested to display a scale bar. In Fig. 3: The authors claim that the two images of E and F are adjacent slices. However, it doesn't look exactly the same at a glance, is there a way to provide more similar slices (same orientation and cancer row) to improve evidence?? Similar situations also appear in Fig. 4 and Fig. 5. 

5. Although there are doubts about whether published case reports or case series constitute research in human subjects and therefore require IRB review (less than 3 cases), given the future trend of protecting the rights of patients, if this study has relevant statements, it is suggested that authors should consider listing.

Reviewer 2 Report

The authors present histopathological data on a rare case of post-polio syndrome (PPS) in a 68-year-old man with a remote history of bulbar and cervical cord involvement by poliomyelitis, precisely review the relevant literature, and contrast the salient histopathologic features that distinguish their case PPS from amyotrophic lateral sclerosis (ALS).

The paper sounds although it is only the case but very valuable.

The histological material is perfectly presented and very convincing.

The state of the art in the Introduction section is presented perfectly, refs. have been selected with high expertise on the topic.

The patient's medical history is presented precisely. Consent was granted for a full postmortem examination. The description of the histological examination and the results is outstanding.

The discussion is interesting and the conclusions are concise.

I found no editorial errors, the citation is in accordance with the MDPI standard.

I would like truly congratulate the authors and recommend this article to be printed in Neurology International.

Author Response

The authors greatly appreciate this reviewer's careful analysis of our case and comparative study of MND cases used to illustrate the neuropathological differences between PPS and primary MND.

Reviewer 3 Report

This is a well-written case study that provides a valuable contribution to the literature on PPS. I have some very minor corrections that are required, see below.

Abstract

“….case of PPS from ALS.”

Figure 1: Please make the following change: “Ventral horn cells”

The following sentence does not make sense. “Furthermore, it is important to bear in mind that within the 15- to 30-year interval between acute poliomyelitis and PPS-associated loss of motor neurons, the patient's age.” – Please revise.

Please correct the following part of this sentence: “In particular, PPMA…” = Should be “PMA”

“AD” – I assume this is Alzheimer’s disease?

Author Response

The authors are most grateful for the careful review and appreciate the attention to detail.  Consequently, the suggested editorial changes that were made (highlighted in green font) improved the manuscript's quality. Thank you.